# The In Vitro Efficacy of Activated Charcoal in Fecal Ceftriaxone Adsorption among Patients Who Received Intravenous Ceftriaxone

**DOI:** 10.3390/antibiotics12010127

**Published:** 2023-01-09

**Authors:** Pattama Torvorapanit, Kornthara Kawang, Pajaree Chariyavilaskul, Stephen J Kerr, Tanittha Chatsuwan, Voraphoj Nilaratanakul

**Affiliations:** 1Division of Infectious Diseases, Department of Medicine, Faculty of Medicine, Chulalongkorn University and King Chulalongkorn Memorial Hospital, Thai Red Cross Society, Bangkok 10330, Thailand; 2Thai Red Cross Emerging Infectious Diseases, King Chulalongkorn Memorial Hospital, Thai Red Cross Society, Bangkok 10330, Thailand; 3Interdisciplinary Program of Biomedical Sciences, Graduate School, Chulalongkorn University, Bangkok 10330, Thailand; 4Department of Pharmacology, Faculty of Medicine, Chulalongkorn University, Bangkok 10330, Thailand; 5Center of Excellence in Clinical Pharmacokinetics and Pharmacogenomics, Faculty of Medicine, Chulalongkorn University, Bangkok 10330, Thailand; 6Biostatistics Excellent Center, Faculty of Medicine, Chulalongkorn University, Bangkok 10330, Thailand; 7Department of Microbiology, Faculty of Medicine, Chulalongkorn, Bangkok 10330, Thailand; 8Center of Excellence in Antimicrobial Resistance and Stewardship, Faculty of Medicine, Chulalongkorn University, Bangkok 10330, Thailand; 9Healthcare-Associated Infection Research Group STAR (Special Task Force for Activating Research), Chulalongkorn University, Bangkok 10330, Thailand; 10Excellence Center for Infectious Diseases, King Chulalongkorn Memorial Hospital, Thai Red Cross Society, Bangkok 10330, Thailand

**Keywords:** activated charcoal, ceftriaxone, gut microbiota

## Abstract

Broad-spectrum antibiotics can kill both pathogens and gut microbiota. Reducing exposure to excess intestinal antibiotics could theoretically protect gut microbiota homeostasis. Recently, engineered charcoals, gel microparticles, and resin beads have demonstrated efficacy in intestinal antibiotic adsorption in animal studies. We report the first in vitro study evaluating human fecal antibiotic adsorption efficacy of conventional activated charcoal (AC). We collected fecal samples from eight patients who received intravenous (IV) ceftriaxone after admission to King Chulalongkorn Memorial Hospital, Thailand, during January–March 2020. Fecal ceftriaxone was measured by indirect competitive enzyme-linked immunoassays. Three different doses of AC were mixed with fecal samples under a specified protocol. The geometric mean reduction in fecal ceftriaxone concentration when mixed with AC 30 mg/g feces was 0.53 (95% CI 0.33–0.85, *p*-value < 0.001), meaning 47% adsorption efficacy. Increased adsorption was found with higher doses, 71% and 87% for AC 150 and 500 mg/g feces, respectively. In conclusion, the usual food-poisoning-care dose of conventional AC, 30 mg/g feces, demonstrated dose-dependent and significant fecal ceftriaxone adsorption. Conventional oral AC might be a pragmatic and inexpensive option for the protection of gut microbiota in patients receiving IV ceftriaxone. However, in vivo studies and microbiome analysis are needed for further evidence.

## 1. Introduction

The widespread use of broad-spectrum antibiotics is associated with inevitable collateral damage. After systemic administration of antibiotics, a fraction of unmetabolized parent compounds or their active metabolites can be excreted into feces and urine. Excess antibiotics in the intestine result in human intestinal microbiota dysbiosis, which commonly leads to colonization with multi-drug-resistant organisms (MDROs) and sometimes later infection [1,2,3,4]. Recent increases in MDROs and *Clostidioides difficile* mortality demonstrate the seriousness of this dysbiosis [4,5]. The colon has the largest, most diverse, and dense range of microorganisms among the human microbiota and helps maintain healthy homeostasis [6]. Changes in the composition of gut microbiota induced by MDRO infection or other mechanisms can affect the metabolic, physiologic, and biochemical profiles of the human host [7]. 

Antibiotic stewardship programs are the most beneficial strategy for protecting gut microbiota from unnecessary broad-spectrum antibiotics, but co-interventions are needed when long-course antibiotics are unavoidable, including gut microbiota restoration and intestinal antibiotic degradation and adsorption. Despite its effectiveness in microbiota restoration, fecal microbiota transplantation (FMT) to correct dysbiosis is infrequently used due to concerns about the safety and hygiene of fecal materials [8,9]. Therefore, minimizing excess colonic antibiotics might be a preferred option to protect gut microbiota. Excess colonic antibiotics can be removed by either degradation, using oral beta-lactamase agents [10,11], or adsorption. Recently published studies in animal models showed some success in protecting gut microbiota from antibiotics using novel synthesized adsorbents, including gel microparticles [12] and resin beads [13].

Activated charcoal (AC) is a well-known adsorbent commonly used in toxicology [14]. The micropore surface of AC can adsorb many substances and drugs [15]. AC was once used as an adsorbent in hemoculture bottles to support bacterial growth, especially when blood samples were drawn after antibiotic administration [16]. In vitro data confirmed the efficacy of AC to adsorb many antibiotics, including tetracycline, quinolone, and penicillin [17]. In 2012, a specialized AC coated with a special plant formula and named DAV131 was engineered to be active only in the colonic environment. DAV131 was successful in preventing beta-lactam-resistant *Klebsiella pneumoniae* colonic colonization in mice after receiving cefotaxime [18]. Subsequently, another novel AC, DAV132, demonstrated significant efficacy in protecting the human gut microbiota after administration of oral moxifloxacin, compared with a control group [19]. However, the production of specialized DAV132, gel microparticles, or resin beads is complicated and expensive. Conventional AC might be a more practical option for widespread and routine use.

Ceftriaxone is one of the most commonly used antibiotics, but its indirect damage to the gut microbiota is overlooked. A 2 g dose of ceftriaxone injected intravenously (IV) reaches an average peak plasma level of 257 mcg/mL at 30 min. Ceftriaxone is eliminated through urine (34–63%) and feces (30%) [20,21,22]. Although data are limited, human fecal ceftriaxone concentrations range widely between 10 and 1000 mcg/mL, 24–48 h after receiving IV ceftriaxone [21]. These levels are high enough to alter gut microbiota, as demonstrated by Brautigam et al. (1988) [3]. Therefore, we decided to evaluate the in vitro adsorption efficacy among three different doses of conventional oral AC in our study to cover various fecal ceftriaxone levels.

## 2. Materials and Methods 

### 2.1. Enrollment

This study was conducted at King Chulalongkorn Memorial Hospital, an academic tertiary-care hospital in Bangkok, Thailand. We collected fecal samples from patients aged ≥18 years, who received at least one intravenous dose of ceftriaxone for any indication, except enteritis or colitis, and had normal bowel movements without laxative use or evacuation. All patients signed written informed consent. We excluded fecal samples from patients exposed to beta-lactam antibiotics in the previous three months. In addition, we collected one null fecal sample from a healthy volunteer who was not exposed to ceftriaxone, who voluntarily signed the consent form. The null fecal sample was included as a control and processed according to the same protocol as the other samples.

The study was approved by the Institutional Review Board of the Faculty of Medicine, Chulalongkorn University, in accordance with the International Guidelines for Human Research Protection, as in the Declaration of Helsinki, the Belmont Report, the CIOMS Guideline, and the International Conference on Harmonization in Good Clinical Practice (IRB no. 335/62). It was also approved by Chulalongkorn University Institutional Biosafety Committee (MDCU-IBC020/2019) and was registered in the Thai Clinical Trials Registry (TCTR20190917003).

### 2.2. Study Design and Sample Size Calculation

This was a preliminary in vitro study to demonstrate the concept of over-the-counter conventional oral-activated charcoal (AC) efficacy in fecal ceftriaxone adsorption among patients receiving intravenous ceftriaxone.

The final sample size was determined using preliminary data from 3 patients with a geometric mean reduction in stool ceftriaxone concentrations at charcoal doses of 30, 150, and 500 mg of 60%, 80%, and 93%, respectively. We assumed the error variance was 1 and there was no within-patient correlation to maximize the sample size. Based on these assumptions, six patients would give 90% power to detect these differences in repeated measures at a two-sided significance level of 5%.

### 2.3. Fecal Sample Collection and Standardization

Fecal samples were collected in sterile plastic containers, immediately stored at 4 °C, and processed within 24 h, as per the following steps.

One gram of fecal sample was mixed and homogenized with 9 mL normal saline (NSS), resulting in 1:10 fecal solutions that were comparable to fluid concentration in the human cecum [23].Fecal solutions were incubated at 70 °C for 30 min to inhibit any residual beta-lactamase enzyme activity that might have been present in some patients, even if they were not exposed to beta-lactam antibiotics in the previous three months [24].The incubated fecal solutions were centrifuged for 5 min at 3000 RCF.Each “fecal medium” supernatant was transferred to a 1.5 mL aliquot tube and frozen at −80 °C for stabilization of ceftriaxone concentrations throughout the study period, in compliance with standard laboratory guidelines.

### 2.4. Fecal Sample Mixing and ‘X’ Fecal Sample Preparation Protocol

We tested the efficacy of adsorption among three different doses of AC: (1) low dose of 3 g per day, which is the recommended conventional AC dose for supportive care of food poisoning; (2) intermediate dose of 15 g per day of engineered DAV132 used in a previous study [19]; and (3) high dose of 50 g per day, which is the maximum dose for adult detoxification. The average adult fecal excretion of approximately 100 g per day was used to calculate the AC/feces ratio. The protocol was set to mix 30 mg, 150 mg, and 500 mg AC with a standardized fecal medium for adsorption efficacy comparison. A mixing environment was planned to mimic the best adsorption of ceftriaxone in the human cecum (Appendix A), so the mixtures of standardized fecal medium and AC were incubated at 37 °C for 3 h in a continuous shaking incubator [25]. The incubated mixture was then centrifuged for 5 min at 3000 RCF. The last supernatants, documented as “X_x_” fecal samples, were pipetted for ceftriaxone-level assays. The baseline standardized fecal medium without the addition of AC was simultaneously incubated in the same way and used as the baseline denominator for adsorption analyses.

### 2.5. Fecal Ceftriaxone Level Measurement

Indirect-competitive enzyme-linked immunosorbent assay (ELISA) was used to measure ceftriaxone concentration in this study. We strictly followed the manufacturer’s instruction for a commercial ELISA kit from Creative Diagnostics^®^ manufacturer, New York, NY, USA (Catalog #DEIABL-QB31^TM^) on a 96-well ELISA plate. The lower limit of detection was 0.5 ng/mL of standard ceftiofur, and the upper limit was 40.5 ng/mL. The ceftriaxone standard curve was generated based on comparable ceftiofur concentrations (Figure 1). Each tested sample was loaded manually into duplicate wells. The results were read by a MULTISKAN^TM^ GO (Thermo Fisher Scientific Incorporated, Waltham, MA, USA) microplate spectrophotometer at 450 nm wavelength.

A standard of 30 ng/mL ceftriaxone in normal saline (NSS) was used as a positive control (ceftriaxone 30 ng/mL), and pure NSS was used as a negative control (ceftriaxone < 0.5 ng/mL). Before analysis of study samples, the incubation environment for the standardized fecal medium and mixing protocol mentioned above was repeatedly tested to ensure it did not alter the ceftriaxone concentrations in the positive or negative controls measured by this ELISA kit. We also tested AC in NSS solution and demonstrated the AC itself did not disturb the assay reactions. 

‘X_x_’ fecal samples were diluted in multiple dilutional titrations to discover the optimal dilution for each ‘X’ sample that was within the detectable limit of this ELISA kit. These dilutional factors were used to calculate the final analysis concentrations before analysis and interpretation.

### 2.6. Statistical Methods

The primary outcome was the average geometric mean reduction (95% CI, *p*-value < 0.05) of the fecal ceftriaxone level by the activated charcoal (AC) in low, intermediate, and high doses, compared to the baseline sample without addition of AC as a reference. Generalized estimating equations with an outcome of natural log (ln) ceftriaxone concentration were used to account for within- and between-patient variability; model coefficients and 95%CI were exponentiated to obtain the geometric mean ratio relative to the baseline sample. We also expressed the adsorption efficacy as percent (%) reduction. Statistical analysis was performed with Stata 16 (Statacorp LLC, College Station, TX, USA).

## 3. Results

Eight prepared ‘X_1_–X_8_’ fecal samples were collected from patients who received IV ceftriaxone and admitted to King Chulalongkorn Memorial Hospital from January to March 2020. Two patients received only one dose, three received two doses, two received two doses, and only one patient received four doses of IV ceftriaxone prior to fecal sample collection. The ‘X_Null_’ fecal sample (Null; control) was prepared from a healthy volunteer who had not received any beta-lactam antibiotics for at least three months. Clinical data and details of IV-ceftriaxone administration prior to fecal sample collection are shown in Table 1.

As previously mentioned, we confirmed the test by adding low-dose AC to a positive control solution (ceftriaxone 30 ng/mL) at room temperature. AC could significantly adsorb ceftriaxone, reducing its level from 30 ng/mL to 6 ng/mL. Furthermore, incubation according to the mixing protocol showed even more ceftriaxone adsorption, resulting in undetectable levels for the ELISA. This supported our development of a system mimicking the human colonic environment in the mixing protocol for adsorption enhancement.

Initially, we tested different brands (A, B, and C) of conventional oral AC marketed in Thailand with only one representative sample, X_3_, due to the limited availability of the ELISA kit. We found that brand A had higher percent adsorption of fecal ceftriaxone compared to brand B at the low dose, while brand C showed a significantly higher percent adsorption of fecal ceftriaxone compared to brand A at the intermediate dose. We decided to select brand C, which had the highest in vitro efficacy in fecal ceftriaxone adsorption for this study (Appendix A).

For optimal dilution of the ‘Xx’ fecal samples, we found that dilution factors of 100× for X_1–2_ and 2000× for X_3–8_ ensured the level of fecal ceftriaxone could be measured within the ELISA range of detection (Appendix A). Fecal ceftriaxone levels at baseline and AC mixing at low, intermediate, and high doses among X_1_-X_8_ and X_Null_ fecal samples are presented in terms of raw measurable data (ng/mL), at the optimal dilution for each “Xx” sample (Table 2).

Absolute fecal ceftriaxone concentration (mcg/mL) of all ‘X’ fecal samples was calculated by multiplication by the dilutional factor for that sample (Figure 2). Baseline samples of X_1_ and X_2_, collected after a single dose of IV ceftriaxone, showed a lower secreted baseline fecal ceftriaxone level (mcg/mL) than X_3–8_. Baseline fecal ceftriaxone levels after one dose of IV ceftriaxone were 1 mcg/mL in X_1_ and nearly undetectable in X_2_. Higher levels of 9, 27, 20, 6, 20, and 120 mcg/mL were detected in X_3–8_, respectively. The mean baseline fecal ceftriaxone level after multiple doses of IV ceftriaxone was 33.67 mcg/mL (±SD 39.25) in our study, 18.67 (±SD 7.41) after two doses (X_3–5_), 13 (±SD 7) after three doses (X_6–7_), and 120 after four doses (X_8_). The baseline levels of fecal ceftriaxone also correlated with the post-IV ceftriaxone period prior to sample collection (Figure 3). This could imply that continuous IV-ceftriaxone use might lead to higher cumulative fecal drug levels.

Absolute levels of fecal ceftriaxone (mcg/mL) after AC mixing declined for all patients (Figure 2). The average geometric mean reductions (GMRs) in fecal ceftriaxone concentrations after AC mixing at different doses, compared to baseline of each Xx sample, are shown in Table 3. Since the 100× dilution of the baseline X_2_ sample was at the borderline of the lower detection limit of the ELISA kit, the effect of AC mixing might be unreliable. In contrast, the raw data of a 2000× dilution of baseline X_8_ were more than the highest detection limit of the ELISA kit, so the effect of AC mixing might be underestimated. Therefore, a sensitivity analysis was performed by excluding these samples, resulting in three datasets: N = 8 (all X_1–8_ samples), N = 7 (exclusion of the X_2_ sample), and N = 6 (exclusion of X_2_ and X_8_). However, the amount of fecal ceftriaxone adsorbed by AC was consistent at all doses among all data sets with effect sizes and 95%CI of similar magnitudes (Table 3 and Figure 4).

Furthermore, stronger reductions in fecal ceftriaxone concentrations were evident with increasing AC doses, with maximum absorption occurring at the highest AC doses (Table 3 and Figure 5).

## 4. Discussion

Activated charcoal (AC) has a high surface area, which facilitates the adsorption of toxic substances; it is commonly used for gastric decontamination. Furthermore, conventional oral AC is also used for supportive care in food poisoning without documented clinical benefit. DAV132, an engineered AC, which functions only in the human colon, could adsorb moxifloxacin that was secreted to the bowel without interfering with plasma drug levels [19]. This suggests that AC may be able to protect the gut microbiota against unavoidable collateral damage from systemic antibiotic use.

Our study is the first preliminary in vitro study demonstrating the efficacy of conventional oral AC, an inexpensive drug, to adsorb excess fecal ceftriaxone. We found that the daily recommended dose of 3 g per day of commercially available oral AC was effective in adsorbing excess ceftriaxone in feces. Prescribing conventional oral AC together with IV ceftriaxone simultaneously might adsorb excess fecal ceftriaxone and alleviate intestinal microbiota dysbiosis.

The cumulative fecal concentration was quite different between single and multiple doses of IV ceftriaxone. There was a trend of higher baseline fecal ceftriaxone levels with increasing numbers of IV doses of ceftriaxone given prior to sample collection. This suggests that longer use of antibiotics is associated with an increased risk of microbiota destruction by excess fecal antibiotics. 

Due to our in vitro study design, we underestimated unexpected situations that arose in some patient samples. The X_2_ and X_6_ samples, collected from patients who had constipation during their illnesses, could be antecedent fecal samples that did not reflect the actual ceftriaxone concentration in their colons. Very low concentrations of fecal ceftriaxone concentrations were observed in both cases, especially the X_2_ sample that was collected after only one dose of IV ceftriaxone. However, our sensitivity analysis, which excluded the X_2_ and X_8_ samples where ceftriaxone levels were below the assay limit of detection, showed effect sizes and 95%CI that were equivalent in magnitude across three datasets: dataset 1 (all X_1–8_), dataset 2 (excluded X_2_), and dataset 3 (excluded X_2_ and X_8_). Higher percentages of fecal ceftriaxone adsorption were also associated with higher doses of AC in admixtures, most likely due to more surface micropores of AC. For AC to be useful in clinical practice, the appropriate AC dose should not burden the patients with a large pill burden during IV ceftriaxone therapy to facilitate patient compliance, but it should be high enough to demonstrate a protective benefit to the gut microbiota. An in vivo study of adsorption efficacy of low-dose, 3 g per day, conventional AC given to healthy volunteers receiving IV ceftriaxone, approximate 50% reduction in intestinal ceftriaxone, would provide further evidence regarding the clinical utility of this strategy in protecting intestinal microbiota and reducing problems associated with MDROs. 

A limitation of our study was the small sample size, but our repeated-measures study had 90% power to detect reductions observed in a preliminary set of samples and we further inflated our sample size to eight. Second, our mixing protocol was designed to mimic conditions in the caecum (Appendix A), but the extent to which this is representative of the entire bowel is unclear. Third, conventional oral AC cannot be used together with broad-spectrum oral antibiotics as it inhibits systemic drug absorption. However, our study has relevance to antibiotics administered parenterally, such as ceftriaxone, which are widely used in many settings. Last, our study needed to find the optimal dilutional factor for each fecal sample due to the narrow detection range of the ELISA kit. This complicated the assay procedure, making it unsuitable for use in routine service. Unfortunately, ceftriaxone-level assessment by high-performance liquid chromatography (HPLC) was not available in Thailand during the study period. However, HPLC protocol can be developed for larger-scale research in the case of applicable AC in the future.

## 5. Conclusions

Our in vitro study demonstrated that conventional AC showed significant efficacy in adsorbing ceftriaxone in fecal samples from patients receiving IV ceftriaxone. This study could help develop a clinical study to demonstrate the clinical benefits in protecting intestinal microbiome homeostasis in the future, and we encourage physicians to consider the importance of gut microbiota protection. Judicious and appropriate use of antibiotics is the most important strategy, but minimizing antibiotic concentrations excreted in feces by activated charcoal (AC) might be an alternate or adjunctive option.

## Figures and Tables

**Figure 1 antibiotics-12-00127-f001:**
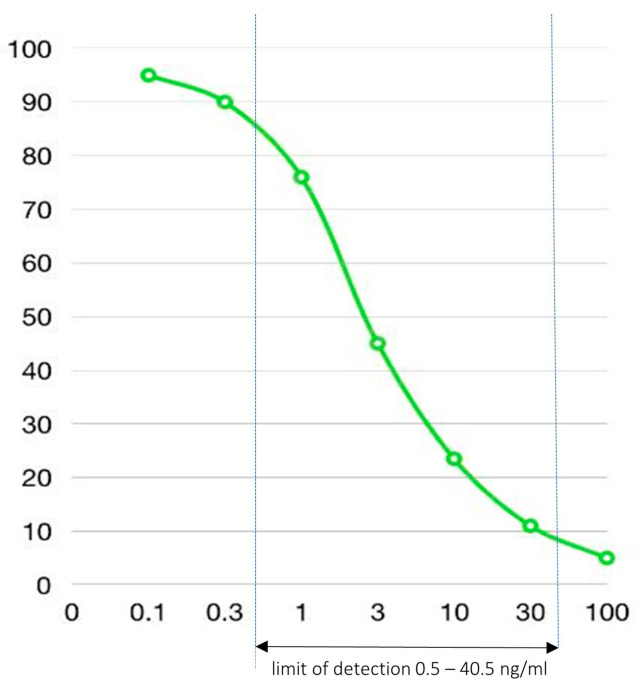
Ceftriaxone standard curve from ELISA kit. *x*-axis = concentration (ng/mL); *y*-axis = optical density absorbance spectrophotometry (%).

**Figure 2 antibiotics-12-00127-f002:**
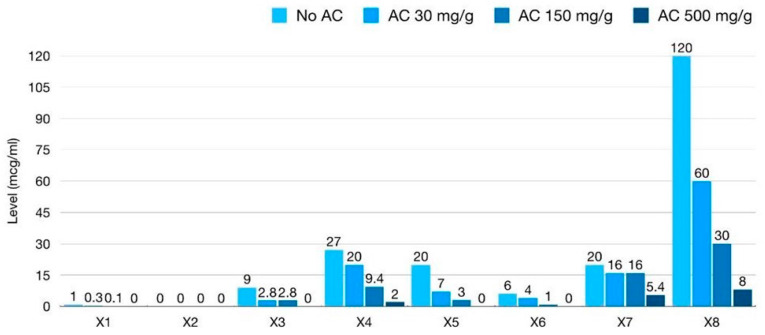
Bar chart comparison of absolute fecal ceftriaxone levels (mcg/mL) at baseline and after mixing with conventional activated charcoal (AC) at different doses: low dose (30 mg/g feces), intermediate dose (150 mg/g feces), and high dose (500 mg/g feces).

**Figure 3 antibiotics-12-00127-f003:**
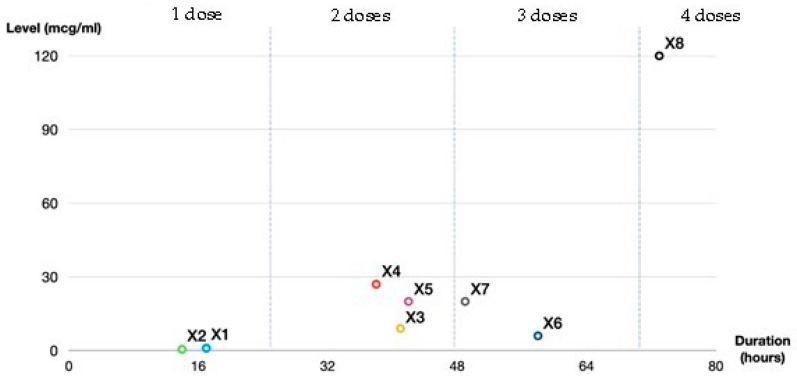
The association of fecal ceftriaxone levels (mcg/mL) and post-IV-ceftriaxone duration prior to fecal sample collection (hours).

**Figure 4 antibiotics-12-00127-f004:**
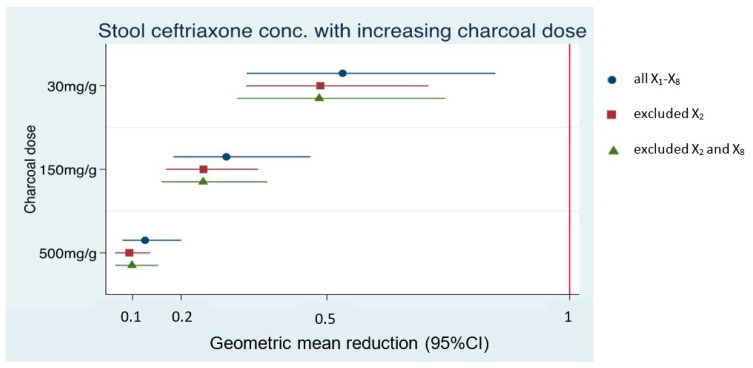
Geometric mean reductions in fecal ceftriaxone concentrations adsorbed by activated charcoal at different doses (low dose 30 mg/g feces, intermediate dose 150 mg/g feces, and high dose 500 mg/g feces) relative to baseline, demonstrated in three datasets: all X_1–8_ (N = 8), excluded X_2_ (N = 7), and excluded X_2_ and X_8_ (N = 6).

**Figure 5 antibiotics-12-00127-f005:**
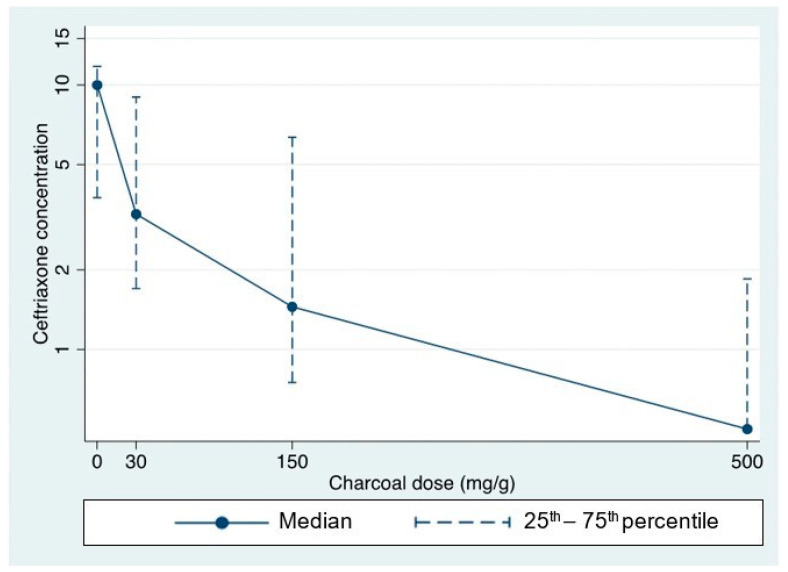
Graph showing decreasing median (IQR) ceftriaxone concentrations (mcg/mL) with increasing activated charcoal doses added; dataset 1 (N = 8, all X_1–8_). Note: *y*-axis is on a logarithmic scale.

**Table 1 antibiotics-12-00127-t001:** Demographic and clinical characteristics of study participants.

X _patient_ Fecal Sample	Demographic Data (Age—Years, Sex)	Primary Diagnosis	Date of IV Ceftriaxone Initiation (mm/dd/yy)	Post 1st IV Ceftriaxone Period Prior to Fecal Sample Collection (hours)	Number of Ceftriaxone Doses Received before Fecal Sample Collection
X_1_	35, female	CBD stone (Elective ERCP)	1/13/2020	17	1
X_2_	69, male	Acute cellulitis	1/13/2020	14	1
X_3_	69, female	Infected wound	1/13/2020	41	2
X_4_	52, male	Acute pyelonephritis	1/15/2020	38	2
X_5_	69, male	Bacterial pneumonia	1/19/2020	42	2
X_6_	79, female	Acute cholangitis	3/2/2020	58	3
X_7_	45, male	Acute pyelonephritis	3/10/2020	49	3
X_8_	89, female	Acute pyelonephritis	3/13/2020	73	4
X_Null_	25, female	Healthy volunteer	NA	NA	0

NA = not applicable.

**Table 2 antibiotics-12-00127-t002:** Fecal ceftriaxone levels (ng/mL) of all ‘X_X_’ samples at their optimal dilution.

X_patient_ Fecal Sample (Optimal Dilution)	Baseline (No AC)	Mixing with Low-Dose AC 30 mg/g Feces	Mixing with Intermediate-Dose AC 150 mg/g Feces	Mixing with High-Dose AC 500 mg/g Feces
X_1_ (100× dilution)	10 ng/mL	3 ng/mL	1 ng/mL	<0.5 ng/mL
X_2_ (100× dilution)	0.5 ^#^ ng/mL	<0.5 ng/mL	<0.5 ng/mL	<0.5 ng/mL
X_3_ (2000× dilution)	4.5 ng/mL	1.4 ng/mL	1.4 ng/mL	<0.5 ng/mL
X_4_ (2000× dilution)	13.5 ng/mL	10 ng/mL	4.7 ng/mL	1 ng/mL
X_5_ (2000× dilution)	10 ng/mL	3.5 ng/mL	1.5 ng/mL	<0.5 ng/mL
X_6_ (2000× dilution)	3 ng/mL	2 ng/mL	0.5 ng/mL	<0.5 ng/mL
X_7_ (2000× dilution)	10 ng/mL	8 ng/mL	8 ng/mL	2.7 ng/mL
X_8_ (2000× dilution)	>40.5 (60) * ng/mL	30 ng/mL	15 ng/mL	4 ng/mL
X_Null_ (100× dilution)	<0.5 ng/mL	<0.5 ng/mL	<0.5 ng/mL	<0.5 ng/mL
X_Null_ (2000× dilution)	<0.5 ng/mL	<0.5 ng/mL	<0.5 ng/mL	<0.5 ng/mL

^#^ Raw data of 100× dilution of X_2_ was borderline at the lowest detection limit of the ELISA kit at 0.5 ng/mL. * Raw data of 2000× dilution of X_8_ was more than the highest detection limit of the ELISA kit. The calculated value (60 ng/mL) was retrieved from the ceftriaxone standard curve (Figure 1).

**Table 3 antibiotics-12-00127-t003:** Geometric mean reduction (GMR), 95%CI and *p*-value, and % adsorption by activated charcoal (AC) at low dose 30 mg/g feces, intermediate dose 150 mg/g feces, and high dose 500 mg/g feces.

**Dataset 1; N = 8 (All X_1–8_)**		
Low dose AC	GMR 0.53 (0.33–0.85), *p*-value = 0.008	47% adsorption
Intermediate dose AC	GMR 0.29 (0.18–0.47), *p*-value < 0.001	71% adsorption
High dose AC	GMR 0.13 (0.08–0.2), *p*-value < 0.001	87% adsorption
**Dataset 2; N = 7 (exclude X_2_)**		
Low dose AC	GMR 0.49 (0.33–0.71), *p*-value < 0.001	51% adsorption
Intermediate dose AC	GMR 0.25 (0.17–0.36), *p*-value < 0.001	75% adsorption
High dose AC	GMR 0.09 (0.06–0.14), *p*-value < 0.001	91% adsorption
**Dataset 3; N = 6 (exclude X_2_ and X_8_)**		
Low dose AC	GMR 0.48 (0.31–0.74), *p*-value = 0.001	52% adsorption
Intermediate dose AC	GMR 0.25 (0.16–0.38), *p*-value < 0.001	75% adsorption
High dose AC	GMR 0.1 (0.06–0.15), *p*-value < 0.001	90% adsorption

## Data Availability

All raw data are available from the corresponding author, V.N., upon reasonable request.

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
