# Peer review of "The In Vitro Efficacy of Activated Charcoal in Fecal Ceftriaxone Adsorption among Patients Who Received Intravenous Ceftriaxone"

_antibiotics, 2023, doi:10.3390/antibiotics12010127_

Round 1

Reviewer 1 Report

The paper: "The in vitro Efficacy of Activated Charcoal in Fecal Ceftriaxone Adsorption among Patients Received Intravenous Ceftriaxone" evaluates the efficacy of conventional activated charcoal (AC) for adsorption of fecal ceftriaxone. Pateints received ceftriaxone prior to fecal collection then levels of fecal ceftriaxone were measured by indirect competitive enzyme-linked immunoassays. The adsorption efficacy was calculated to be 47%. Higher doses of AC produced higher adsorption efficacy. They therefore conclude that the conventional AC, at the usual food poisoning care dose of 30g/g feces, could adsorb  fecal ceftriaxone and therefore represent a pragmatic and inexpensive option for the protection of the gut microflora.

Abstract – the abstract is disjoined and not clearly written.

Throughout the text: "gut microflora" – the correct term is microbiome not microflora.

Line 25: "Adsorption of excess intestinal antibiotics might protect gut microflora" – It is not clear enough as a sentence and is very disjoined from the sentence above it.

Line 45: "collateral problems," – I would think either collateral damage or more problems are more grammatically correct.

Line 63 " the colon is the largest part" – part is not the word you should be using here

Introduction – The first two paragraphs are very badly written. I think that the problem here is lack of focus. Introducing an array of bacteria as examples, mostly without the reason why they are chosen as examples being rather obscure, makes this a less than ideal introduction, specifically one that is not properly introducing your research. Why do you speak of clostridium and antibiotic stewardship in the same paragraph? Why is ribaxamase” important here? The logic is not clear, or not clearly written. Paragraph 3 is, by contrast, much better and uses the correct term "microbiota".  

Line 95: I wish you move the line "Ceftriaxone is one of the most used antibiotics, but its collateral damage to the gut microflora is overlooked" to the start of the paragraph as it is a better starting point to the paragraph as a whole.

Materials and Methods – I am worried that a single null sample is not enough for good comparison, especially considering the low number of samples in this study.

Materials and Methods – "As mentioned before, we confirmed the test by adding low-dose AC to a positive control solution (ceftriaxone 30 ng/ml) at room temperature. AC could significantly adsorb ceftriaxone, reducing its level from 30 ng/ml to 6 ng/ml. Furthermore, incubation under the mixing protocol showed even more ceftriaxone adsorption, resulting in an undetectable level by the ELISA. This supported us in developing a mimic of the human colonic environment in the mixing protocol for adsorption enhancement" – are you sure that these conditions are indeed in line with the conditions of the human colon? Was the incubation in 37 degrees? Was it done in darkness? Are you sure that the differences in the colonic and in vitro environment were sufficiently measured and taken into account? If so it needs to be stated more clearly in this section.

Line 333: " Feasible AC doses will be discussed in our secondary research in healthy volunteers." – why state here what you are planning to do next? It is redundant

Reviewer 2 Report

1.       ELISA kit is sufficient for detection of ceftriaxone?? Is there any other technique for the detection of ceftriaxone??

2.       Add some latest references in the manuscript.

Author Response

Response to Reviewer 2's comments

Point 1. ELISA kit is sufficient for detection of ceftriaxone?? Is there any other technique for the detection of ceftriaxone??

Response 1. Unfortunately, fecal ceftriaxone level assessment by high-performance liquid chromatography (HPLC) was not available in Thailand during the study period. We documented this limitation in our manuscript. To mitigate the limitations of ELISA, we undertook dilution factor discovery for each sample to ensure the concentrations fitted the ELISA kit range of detection, and all tests at every step were conducted in duplicate. These ELISA results should properly represent the trend of in-vitro efficacy of activated charcoal on fecal ceftriaxone adsorption. – page 5, line 231-234 and page 14, line 428-435

Point 2. Add some latest references in the manuscript.

Response 2. We appreciate your recommendation. Additional relevant references have been updated and incorporated into our revised manuscript.

Round 2

Reviewer 1 Report

I have two minor comments:

Abstract line 25 - "despite not aiming to treat colitis" – why is this important? Colitis is not discussed in the paper.

 Line 53: "but co-interventions are needed 53 when long-course antibiotics are unavoidable" – please mention what co-interventions you are referring to

Author Response

Response to Reviewer 1's comments

Point 1. Abstract line 25 - "despite not aiming to treat colitis" – why is this important? Colitis is not discussed in the paper.

Response 1.

  • We agree that this phrase is not necessary and should be removed. - page 1, line 25
  • Originally, this is to emphasize that the colon is usually not the target site for antibiotic actions. Therefore, if we can get rid of residual antibiotics in the colon, we can avoid killing colonic microbiota without compromising antibiotic treatment outcomes against infections at other sites.

Point 2. Line 53: "but co-interventions are needed 53 when long-course antibiotics are unavoidable" – please mention what co-interventions you are referring to

Response 2. These co-interventions are mentioned later in the same paragraph. We also slightly revised the phrases to improve clarity. - page 2, line 56-63

Notes:

  • Responses are in red
  • Pages and lines mentioned in this reviewer repsonse, were specified to the revised manuscript.